# The Adjustment Strategy of Venus Flytrap Photosynthetic Apparatus to UV-A Radiation

**DOI:** 10.3390/cells11193030

**Published:** 2022-09-27

**Authors:** Karolina Miernicka, Barbara Tokarz, Wojciech Makowski, Stanisław Mazur, Rafał Banasiuk, Krzysztof M. Tokarz

**Affiliations:** 1Department of Botany, Physiology and Plant Protection, Faculty of Biotechnology and Horticulture, University of Agriculture in Krakow, Al. 29 Listopada 54, 31-425 Kraków, Poland; 2Institute of Biotechnology and Molecular Medicine, Kampinoska 25, 80-180 Gdansk, Poland

**Keywords:** abiotic stress, *Dionaea muscipula* J. Ellis, photosynthesis, short-wave radiation

## Abstract

The objective of this study was to investigate the response of the photosynthetic apparatus of the Venus flytrap (*Dionaea muscipula* J. Ellis) to UV-A radiation stress as well as the role of selected secondary metabolites in this process. Plants were subjected to 24 h UV-A treatment. Subsequently, chl *a* fluorescence and gas exchange were measured in living plants. On the collected material, analyses of the photosynthetic pigments and photosynthetic apparatus proteins content, as well as the contents and activity of selected antioxidants, were performed. Measurements and analyses were carried out immediately after the stress treatment (UV plants) and another 24 h after the termination of UV-A exposure (recovery plants). UV plants showed no changes in the structure and function of their photosynthetic apparatus and increased contents and activities of some antioxidants, which led to efficient CO_2_ carboxylation, while, in recovery plants, a disruption of electron flow was observed, resulting in lower photosynthesis efficiency. Our results revealed that *D. muscipula* plants underwent two phases of adjustment to UV-A radiation. The first was a regulatory phase related to the exploitation of available mechanisms to prevent the over-reduction of PSII RC. In addition, UV plants increased the accumulation of plumbagin as a potential component of a protective mechanism against the disruption of redox homeostasis. The second was an acclimatization phase initiated after the running down of the regulatory process and decrease in photosynthesis efficiency.

## 1. Introduction

UV radiation constitutes 6.3% of the solar radiation reaching the Earth’s surface and 95% of total UV radiation is UV-A radiation. Although UV-A (315–400 nm) is identified as the least hazardous type of UV radiation, with its longer wavelength, it penetrates deeper into tissues than UV-B (280–315 nm) and UV-C (100–280 nm), affecting deeper processes, including photosynthesis [1,2]. One of the direct effects of UV-A radiation on photosynthesis is the photoinhibition of photosystem II (PSII). Photoinhibition is defined as the dysfunction of PSII caused by photodamage, due to excessive light both in terms of duration and quantity as well as light quality, especially short-wavelength contents [3]. UV-A radiation leads to the disruption of electron transport at the level of the oxygen-evolving complex (OEC) and plastoquinone B binding site (Q_B_) [4]. The main component of the electron transport chain damaged directly by UV-A is Mn_4_O_5_Ca molecule in OEC [5]. OEC damage increases the generation of reactive oxygen species (ROS), resulting in disturbances of D1 and D2 proteins in the PSII reaction center (RC) and electron binding sites (Q_A_ and Q_B_). Plastoquinone, a membrane electron transporter that absorbs radiation in the UV-A range, is also damaged [6], whereas photosystem I (PSI) is believed to be significantly more UV-A-resistant than PSII [7]. UV-A radiation may also be detrimental to key photosynthetic enzymes, such as 1,5-bisphophoribulose carboxylase/oxygenase (Rubisco) [7], ATP-ase [8] and violaxanthin de-epoxidase [9,10]. Such damage results from the absorption of longer wavelengths (315–400 nm) of UV radiation by amino acids such as tryptophan, tyrosine and cysteine, leading to their destruction and the inactivation of enzymes [11]. These mechanisms may be also responsible for damaging PSII and PSI protein subunits [12]. On the other hand, UV-A radiation can also positively affect photosynthesis. It increases photosynthesis efficiency in marine phytoplankton [13], makroalgae *Gracilaria lemaneiformis* [14] and especially in plants such as *Liquidambar styraciflua* and *Acer rubrum* [15], *Sorghum bicolor* L. [16], *Pimelea ligustrina* [17] and *Lathyrus sativus* [18]. This phenomenon is explained by three different hypotheses. First, both chlorophylls (Chl) and carotenoids (Car), components of the photosynthetic antenna, exhibit a low absorption minimum in the region corresponding to UV-A radiation and can transfer energy directly to the chlorophyll in RC PSII, causing its excitation [17,19]. Second, the absorption of energy originating from the green/blue fluorescence of phenolic compounds induced by UV-A radiation can occur [19,20]. Third, UV-A can directly stimulate the opening of the stomata, which is known to occur after blue light irradiation [20,21]. Furthermore, the lack of limitation on the acceptor side of PSI, including the undisturbed synthesis of assimilatory power, intensive and efficient carboxylation of CO_2_ (dark phase) and/or efficient functioning of alternative electron transport pathways, reduces the negative effects of UV-A radiation on the photosynthetic apparatus [18].

As photosynthetic process may regulate secondary metabolism, its alterations induced by UV-A radiation would affect the biosynthesis of secondary compounds. The activation shikimic acid or acetate polimalonate pathway, leading to the synthesis of phenolic compounds, may occur through the absorption of short-wave radiation (such as UV-A or UV-B) by photoreceptors [22]. Another way of stimulating the accumulation of phenolic compounds may be through the proline biosynthesis induced in response to stress. Proline increases the ratio of NADP^+^/NADPH and NADP^+^ is a cofactor of the enzyme in the pentose phosphate pathway, leading to the enhancement of phenol synthesis [22]. Phenolic compounds may be also synthesized in pathways depending on phytohormones, such as jasmonate or salicylate. The regulation of these pathways and thus the induction of secondary metabolite biosynthesis is connected to: blue or red light action, increased light intensity or the presence of products formed from the degradation of photosynthesis apparatus caused by abiotic factors (such as UV-A radiation) [23,24]. The irradiation-induced changes in the pathways, leading to the synthesis of phenols, suggest their role as screening compounds protecting photosynthetic apparatus from ROS generated by excess light energy or UV radiation [25,26].

One of the plants exposed to variable UV-A radiation in its natural habitat is the Venus flytrap (*Dionaea muscipula* J. Ellis) [27], which can be found in sun-exposed, open habitats. This monotypic species of Droseraceae is endemic to North and South Carolina (USA) and is a carnivorous plant that captures small prey (insects, spiders, etc.) and uses it as a supplementary nutrient resource [28]. Moreover, research initiated in the 1980s [29] evidenced that extracts of Venus flytrap could be used to treatment of tumors due to oncolytic, antiproliferative and immunostimulating properties [30]. Successive studies revealed that these properties resulted from the contents of many valuable secondary metabolites, especially phenolic compounds, including naphthoquinones, especially plumbagin [24,26,28,31,32]. Plumbagin (5-hydroxy -2-methyl-1,4-naphthoquinone) reveals cytotoxic activity against certain microorganisms; hence, it prevents bacteria and fungi from developing on the prey’s surface in traps of carnivorous plants. Furthermore, plumbagin exhibits antioxidant [33], antibacterial [34], anti-inflammatory [35], antifungal [36] and anticancer activities [37]. The high content of phenols in these plants raises the question of whether and how they participate in the mechanism protecting the photosynthetic apparatus against UV-A radiation.

The objective of this study was to evaluate the adjustment strategy of the photosynthetic apparatus of *Dionaea muscipula* to UV-A radiation stress. Additionally, we examined the relationship of selected secondary metabolites with the response of the photosynthetic apparatus under these conditions. We hypothesized that the short-term application of UV-A radiation would enhance the efficiency of the photosynthesis and synthesis of secondary metabolites, especially naphthoquinones, in *Dionaea muscipula* plants.

Our results revealed that *D. muscipula* plants underwent two phases of adjustment to UV-A radiation. The first was a regulatory phase related to the exploitation of available mechanisms to prevent the over-reduction of PSII RC. In addition, UV plants increased the accumulation of plumbagin as a potential component of a protective mechanism against the disruption of redox homeostasis. The second was an acclimatization phase initiated after the running down of the regulatory process and decrease in photosynthesis efficiency.

## 2. Materials and Methods

### 2.1. Plant Material, Acclimatization and Experimental Conditions

The research material comprised *Dionaea muscipula* J. Ellis plants from established in vitro cultures. Plants were cultivated on ½ MS [38] medium with 30 g·L^−1^ sucrose and pH 5.5, solidified with 8 g·L^−1^ agar. Plants were cultured at 23 ± 1 °C, under 16/8 h light photoperiod of 120 μmol·m^−2^·s^−1^ photosynthetic photon flux density (PPFD). The plants were subcultured in 60-day intervals.

In vitro plants were transplanted to pots with sand and peat in a 1:1 ratio. Plants were acclimated in Sanyo Environmental Test Chamber MLR-351H at 23 ± 1 °C, under 16/8 h light photoperiod of 100 μmol·m^−2^·s^−1^ PPFD and air humidity of 90%. Humidity was reduced every 4 days by 10% until it reached a value of 50%. After two weeks, the light intensity in the chamber was increased to 170 μmol·m^−2^·s^−1^ PPFD. After four weeks, plants were placed in a growing chamber at 23 ± 1 °C, under 16/8 h light photoperiod of 290 μmol·m^−2^ s^−1^ PPFD, and air humidity of 30–40% and acclimated for one week. After acclimatization, plants were treated with UV-A radiation of 50 μmol·m^−2^·s^−1^ PPFD for 24 h. During the light period (16 h), fluorescent light was supplemented with UV-A radiation. Other conditions were the same as described for the last week of acclimatization in the growing chamber. Control plants were grown in the same conditions excluding UV-A radiation. Plants were watered with distilled water every day during acclimatization and the experiment. Chlorophyll *a* fluorescence and gas-exchange measurements were conducted on plant leaves immediately after the termination of the 24 h exposure to UV-A treatment, both in control plants and UV-A-treated plants, refered to as UV plants in the following. Additionally, measurements were taken in UV-A-treated plants after 24 h after the termination of the 24 h exposure to UV-A, the so-called recovery plants (Figure 1). Plant material for the other analyses was also collected on the same dates. Leaves (without trap part) were collected and freeze-dried for 48 h.

### 2.2. Determination of Dry Weight Content

The experimental treatment leaves (without trap part) of 5 plants from each treatment were collected. To calculate the percentage of dry weight content, collected material was weighed, dried for 48 h (in a freeze-dryer) and reweighed.

### 2.3. Estimation of Photosynthetic Apparatus Response

#### 2.3.1. Photosynthetic Pigments Content Estimation

The contents of chlorophyll *a*, chlorophyll *b* and carotenoids was determined according to Lichtenhalter [39] with modifications. In total, 10 mg of dry tissue was extracted in 1 mL of 80% acetone and centrifuged for 15 min (25,155× *g*, 4 °C). Extraction was repeated twice in 0.5 mL of 80% acetone. The absorbance of the extract was measured at 663 nm (Chl *a*), 646 nm (Chl *b*) and 470 nm (Car). The contents of photosynthetic pigments was calculated using Wellburn’s formulas [40] and expressed as milligrams of each pigment per 1 g of DW.

#### 2.3.2. Chlorophyll a Fluorescence Measurement

Chlorophyll *a* fluorescence measurements were conducted on leaves of 10 plants of each combination. Before measurement, plants were subjected to dark conditions for 20 min. Chlorophyll *a* fast kinetic curve was measured with a Handy-PEA spectrofluorometer (Hansatech, King’s Lynn, UK) using standard procedures. After this, dark-adapted plants were subjected to a saturating flash of light at 2000 μmol·m^−2^·s^−1^ intensity for 800 ms. Selected photosynthetic parameters were calculated using the formulas of Strasser et al. [41], Jiang et al. [42], Kalaji et al. [43] and Goltsev et al. [44]. These were: F_0_ (minimum fluorescence, when all PSII reaction centers (RCs) are open); F_m_ (maximum fluorescence, when all PSII reaction centers are closed); F_V_ (variable fluorescence; F_m_ − F_0_); F_V_/F_m_ (maximum quantum yield of PSII; (F_m_ − F_0_)/F_m_); F_V_/F_0_ (activity of the water-splitting complex on the donor side of the PSII; (F_m_ − F_0_)/F_0_); area (activity of the water-splitting complex on the donor side of the PSII; (F_m_ − F_0_)/F_0_); V_J_ (relative variable fluorescence at 2 ms (J-step);V_J_ = (F_2ms_ − F_0_)/(F_m_ − F_0_)); V_I_ (relative variable fluorescence at 30 ms (I-step);V_I_ = (F_30ms_ − F_0_)/(F_m_ − F_0_)); Sm (normalized total complementary area above the OJIP transient (reflecting multiple-turnover *Q_A_* reduction events) or total electron carriers r RC; S_m_ = Area/(F_m_ − F_0_)); φ_Po_ (maximum quantum yield of primary photochemistry at t = 0; φ_Po_ = 1 − F_0_/F_m_ = F_V_/F_m_); φ_Eo_ (quantum yield for electron transport at t = 0; φ_Eo_ = (F_V_/F_m_)(1 − V_J_*)*); ψ_Eo_ (probability (at time 0) that trapped exciton moves an electron into the electron transport chain beyond; ψ_Eo_ = 1 − V_J_); ρ_Ro_ (efficiency with which a trapped exciton can move an electron into the electron transport chain from *Q_A_*^−^ to the PSI and electron acceptors; ρ_Ro_ = ψ_Eo_δ_Ro_ = (1 − V_J_)(1 − V_I_)/(1 − V_J_)); δ_Ro_ (efficiency with which an electron can move from the reduced intersystem electron acceptors to the PSI end electron acceptors; δ_Ro_ = RE_o_/ET_o_ = (1 − V_I_)/(1 − V_J_)); φ_Ro_ (quantum yield for the reduction of end acceptors of PSI per photon absorbed; φ_Ro_ = RE_o_/ABS = φ_Po_ψ_Eo_δ_Ro_); ABS/RC (absorption flux per RC;ABS/RC = Mo/V_J_ = 4(F_300μs_ − F_0_)/(F_m_ − F_0_)/V_J_); TR_o_/RC (trapped energy flux per RC at t = 0;TR_o_/RC = Mo/V_J_); ET_o_/RC (electron transport flux per RC at t = 0; ET_o_/RC = (Mo/V_J_)ψ_Eo_); DI_o_/RC (dissipated energy flux per RC at t = 0; DI_o_/RC = ABS/RC − TR_o_/RC); RC/CS_o_ (amount of active PSII RCs per CS at t = 0; RC/CS_o_ = φ_Po_(ABS/CS_o_)(V_J_/Mo)); TR_o_/CS_o_ (trapped energy flux per CS at t = 0; TR_o_/CS_o_ = (ABS/CS_o_)φ_Po_); ET_o_/CS_o_ (electron transport flux per CS at t = 0; ET_o_/CS_o_ = (ABS/CS_o_)φ_Eo_); DI_o_/CS_o_ (dissipated energy flux per CS at t = 0; DI_o_/CS_o_ = ABS CS_o_ − TR_o_/CS_o_). After recording, the curves were interpreted using the fluorometer manufacturer’s soft-ware (PEA-Plus).

#### 2.3.3. Gas-Exchange Measurement

Gas-exchange measurements were conducted using the gas-exchange system LCpro-SD with a measuring chamber, LCP010/AL (ADC BioScientific Ltd., Hoddesdon, UK), on leaves of Venus flytrap. Net photosynthesis (Pn) was measured under CO_2_-saturated conditions (650 µmol·mol^−1^). Conditions inside the cuvette were set at: 300 mol·s^−1^ of air flow with 50–55% relative humidity, and organ temperature at 25 °C. Measurements were performed under a red light intensity of 100 mol (quanta) m^−2^·s^−1^. The stomatal conductance (Gs), transpiration rate (E) and intercellular concentration of CO_2_ (Ci) were also measured. Photosynthetic light–response curves were created for corresponding plants subjected to net photosynthesis, for a gradual reduction in the photosynthetic active radiation (PAR), ranging from 2000 to 0 µmol (quanta)·m^−2^·s^−1^ (in 0, 20, 50, 100, 300, 500, 1000, 1500 and 2000 µmol (quanta)·m^−2^·s^−1^ steps). Before recording data points, leaves were adapted to each light intensity for 20, 5, 5, 5, 5, 5, 5, 5 and 5 min, respectively.

#### 2.3.4. Protein Content Determination

To determine the occurrence and levels of selected proteins, the Western Blot technique was used. The tissues were extracted in protein extraction buffer (100 mM Tris-HCl, pH 8.0, 10% sucrose, 0.2% β-mercaptoethanol and 2% PVPP) according to Laureau et al. [45] with modifications. The concentration of protein was estimated according to Bradford [46]. Calibration curve was prepared with BSA as a standard. SDS-PAGE electrophoresis was performed on 12% polyacrylamide gels (BioRad) at 4 °C at 30 mA for 15 min and 20 mA for 90 min using a vertical gel electrophoresis system (Mini-PROTEAN^®^ Tetra Vertical Electrophoresis Cell, Bio-Rad, Hercules, CA, USA).

The electroblotting of protein from polyacrylamide gels on polyvinylidene fluoride membrane (PVPD) (BioRad) was performed with a semidry electroblotter (Trans-Blot SD Semi-Dry Transfer Cell, Biorad, CA, USA). Transfer buffer composed of 48 mM Tris (pH 9.2), 39 mM glycine, 20% methanol, and 1.3 mM SDS was used for electron transfer. Other parameters of transfer were: room temperature, 10 V (limiting parameter) and 400 mA. Transfer was performed for 30 min. TBST buffer containing 3% dry milk was used to block membranes at room temperature for 2 h. The membranes were then incubated with rabbit primary antibody (Ab) against ascorbate peroxidase (APX, AS08 368, Agrisera, Vinnas, Sweden), D1 protein of PSII (PsbA, AS05 084, Agrisera), CP47 protein of PSII (PsbB, AS04 038, Agrisera), CP43 protein of PSII (PsbC, AS11 1787, Agrisera) and 33 kDa of the oxygen-evolving complex (OEC) of PSII (anti-protein) (PsbO, AS06 142-33, Agrisera). Next, membranes were washed using TBST buffer. Then, blots were probed with horseradish peroxidase-conjugated anti-rabbit secondary antibody (HRP, AS09 602, Agrisera) at a dilution of 1:10,000 in TBST buffer with 1% dry milk for 1.5 h. Following washing with TBST buffer, the solution of 5-bromo-4-chloro-3-indolyl phosphate (BCIP) and nitro blue tetrazolium (NBT) was used to detect the antigen–antibody complexes. The solution was made in a buffer containing 100 mM Tris (pH 9.5), 100 mM NaCl and 5 mM MgCl_2_. Subsequently, the membranes were digitally scanned using the Epson Perfection V750 Pro scanner. The digitized membranes were analyzed densitometrically using ImageJ software (version 1.53k, open-source software, NIH, Bethesda, MD, USA). The content of each protein is presented as arbitrary units, referring to the area under the curve. The mean area value for control on each gel, expressed as 1, was used to calculate relative area values.

### 2.4. Estimation of Sugar Content

Soluble and insoluble sugar contents were determined using the anthrone reagent method [47]. Firstly, 1 mL Milli-Q-ultrapure water (Millipore Direct system Q3) was used to extract 10 mg of dry tissue overnight. The samples were centrifuged for 10 min (25,155× *g*, room temperature) and supernatants were collected. The aqueous supernatant was used for the determination of soluble sugars. After aqueous extraction, the pellet was resuspended in 0.1 M H_2_SO_4_ (0.5 mL) and heated at 80 °C for 60 min. The acid supernatant was used for the determination of insoluble sugars. Extracts (aqueous or acid) were mixed with anthrone reagent solution (1 g anthrone in 500 mL 72% H_2_SO_4_) and incubated at 95 °C for 15 min. The reaction was terminated on ice. The absorbance (630 nm) of the samples was measured on a Genesys 10 VIS spectrophotometer (Thermo-Fischer Scientific, Waltham, MA, USA) at room temperature. The content of sugars (mg·g^−1^ dw) was calculated from a glucose calibration curve.

### 2.5. Estimation of Lipid Peroxidation Level using Malondialdehyde (MDA) Content

MDA content was estimated according to Dhindsa et al.’s [48] method with modifications. In total, 10 mg of dry tissue was homogenized in 1.2 mL of 0.1% trichloroacetic acid (TCA) at 4 °C. Homogenates were centrifuged for 15 min (25,155× *g*, 4 °C). Then, supernatant (0.5 mL) and 0.5% thiobarbituric acid (TBA) in 20% TCA solution (0.5 mL) were mixed. Reaction mixtures were incubated for 30 min at 95 °C, terminated on ice and centrifuged for 10 min (25,155× *g*, 4 °C). The absorbance of the samples was measured at 532 nm and 600 nm wavelengths. MDA content was calculated according to Dhindsa et al. [48] using the MDA extinction coefficient (ε = 155 mM·cm^−1^). The value of absorbance for the reaction mixture at 532 nm was reduced by the correction value obtained at 600 nm. The results are expressed as nM of MDA per 1 g of DW.

### 2.6. Estimation of Antioxidant Enzymes Activity

#### Peroxidase (POD), Superoxide Dismutase (SOD) and Catalase (CAT) Activity Estimation

The extraction of tissue for POD and SOD was conducted according to Miszalski et al. [49]. In total, 20 mg of dry tissue was extracted in 2 mL of extraction buffer (10 mL of phosphate buffer of pH 7.5, 10 μL 1 M DTT, 11 mg EDTA, 0.2 g PVPP, 100 μL Triton X-100 and 1 mM PMSF (0.0174 g·100 mL) dissolved in few drops of alcohol). Samples were centrifuged for 15 min (25,155× *g*, 4 °C). SOD activity was assessed spectrophotometrically according to Hwang et al. [50] with Wiszniewska et al.’s [51] modifications. The reaction mixtures were prepared by mixing: 2.15 mL of 100 mM phosphate buffer, pH 7.8, 0.2 mL 55 mM methionine, 0.4 mL 0.75 mM nitrotetrazole blue (NBT), 0.2 mL 0.1 mM riboflavin and 50 μL protein extract. The reaction was started by putting the samples under a 40 W lamp at room temperature. The absorbance of mixtures was measured 5 and 10 min after starting the reaction at 560 nm. Samples without extract were the control, where the activity was 100%. Activity is expressed as U per 1 g of DW, where U means the inhibition of the reaction by 50% in relation to the control. POD activity was estimated using the spectrophotometric method of Lűck [52]. A total of 1 mL of diluted extract was mixed with sodium-phosphate buffer (1 mL), pH 6.2 and 0.1 mL of pPD and 0.2 mL of H_2_O_2_. The absorbance of samples was measured at 485 nm. The activity of POD is expressed as U per minute per 1 g of DW, where 1U means an increase in absorbance by 0.1. To estimate CAT activity, tissue was extracted in 1 mL of phosphate buffer (pH 7.0) and centrifuged for 10 min (25,155× *g*, 4 °C). The reaction mixture consisted of 0.2 mL of the extract, 1.8 mL of phosphate buffer and 1 mL of 0.3% H_2_O_2_ solution in phosphate buffer. The absorbance of the mixture was measured for 4 min at 1 min intervals at 240 nm [53]. Catalase activity is expressed in units as the amount of enzyme decomposing 1 μmol H_2_O_2_ in 1 min.

### 2.7. Estimation of Low-Molecular-Weight Antioxidant Contents and Activities

#### 2.7.1. Phenolic Compound Content Estimation

##### Total Phenolic Compounds

The contents of total phenolic compounds was estimated using Folin–Ciocalteu’s method according to Swain and Hillis [54]. A total of 10 mg of dry tissue was homogenized in 1 mL of 80% methanol at 4 °C and centrifuged for 15 min (25,155× *g*, 4 °C). Then, 0.2 mL of Folin–Ciocalteu reagent and 1.6 mL of 5% Na_2_CO_3_ was added to 1.0 mL of the 25 times diluted extract. Samples were incubated for 20 min in 40 °C. A Double-Beam U-2900 spectrophotometer (Hitachi High-Technologies Corp, Tokyo, Japan) was used to measure the absorbance of samples at 740 nm. A calibration curve was prepared with the usage of chlorogenic acid as a standard. The result is expressed as milligrams of chlorogenic acid per 1 g of dry weight (DW).

##### Phenolic Compound Groups

The estimation of phenylpropanoid, flavonoid and anthocyanin contents was conducted according to Fukumoto and Mazza [55] with modifications. Ten milligrams of dry tissue was homogenized in 1 mL of 80% methanol at 4 °C and centrifuged for 15 min (25,155× *g*, 4 °C). To 0.125 mL extract was added 0.125 mL 0.1% HCl in 96% EtOH and 2.275 mL 2% HCl in H_2_O. Mixtures were kept for 20 min in the dark at room temperature. Subsequently, absorbance was measured at 320 nm (phenylpropanoids), 360 nm (flavonoids) and 520 nm (anthocyanins). The results were calculated on the basis of calibration curves prepared with caffeic acid (standard for Phe), quercetin (standard for Fla) and cyaniding (standard for Ant) and expressed as mg per 1 g of DW.

##### Specific Phenolic Compounds

Tetrahydrofuran (THF) extracts for the chromatographic analysis of plumbagin were prepared according to Tokarz et al. [26]. Ten milligrams of dry tissue was extracted in 0.6 mL of MiliQ water and 0.6 mL of THF (C_4_H_8_O) by shaking (20 min) and sonification (30 min). After the addition of NaCl (200 mg), samples were shaken for another 20 min. To extract other phenolic compounds (caffeic acid, hyperoside, ellagic acid and myricetin), 20 mg of dry tissue was extracted in 100% methanol (2 mL), at 4 °C. Next, samples were sonicated for 30 min. Samples were centrifuged for 15 min (25,155× *g*, 4 °C) and supernatants were collected for chromatographic analysis. A chromatographic estimation of the phenol contents was conducted using Beckman System Gold chromatograph equipped with a variable-wavelength detector Thermo Separations Spectra and injection valve Rheodyne 6-way. The stationary phase constituted the Agilent XDB-C18 column (4.6 × 50 mm, 1.8 µm). The volume of the sample was 10 µL and flow rate was 1 mL·1 min^−1^. The mobile phase consisted of methanol (as eluent A) and water (as eluent B). Separation was conducted in isocratic elution conditions (60% of eluent A and 40% of eluent B) at room temperature for 5 min. Analysis was conducted at a wavelength of 254 nm. To estimate the content of phenolic compounds, a 4-point, 3-degree standard curve was used [24].

#### 2.7.2. Estimation of Total Glutathione Content

Total glutathione (reduced glutathione (GSH) + glutathione disulfide (GSSG)) content was estimated according to Queval and Noctor [56]. Ten milligrams of dry tissue was extracted in 1 mL of 0.5 M HCl at 4 °C and centrifuged for 15 min (25,155× *g*, 4 °C). The pH of 400 μL of extract was neutralized with 0.2 M NaOH or 0.5 M HCl until it reached a value between 5.5 and 6.0. Then, 30 μL of neutralized extract, 300 μL of buffer with EDTA (pH 7.5), 30 μL of 10 mM NADPH, 30 μL of 12 mM DTNB and 180 μL of MiliQ water were mixed. To each reaction mixture was added 30 μL of glutathione reductase (GR) (20 U·μL^−1^). The absorbance of mixtures was measured at 412 nm 1 min and 2 min after the addition of GR. The results are expressed as milligrams of GSSH + GSSG per 1 g of DW.

### 2.8. Statistical Analysis

Statistical analyses were conducted using STATISTICA 13.1 (StatSoft Inc., Tulsa, OK, USA). The results were analyzed by one-way analysis of variance (ANOVA), and the significance of differences between the arithmetical means was tested using Duncan’s post hoc test at *p* ≤ 0.05. Spectrophotometric and HPLC estimations were made in five and three replications, respectively. Chl *a* fluorescence measurements were made in ten replications. Gas-exchange measurements and electrophoresis were conducted in five and three replications, respectively.

## 3. Results

### 3.1. Growth of D. Muscipula Plants under UV-A treatment

After 24 h of continuous UV-A treatment, no changes in plant morphology were observed compared to control plants, both in plants directly after UV-A treatment (hereinafter referred to as UV plants) and 24 h after termination of 24 h exposure to UV-A treatment (hereinafter referred to as recovery plants) (Figure 2a), whereas a statistically significant reduction in plant dry weight content was observed in recovery plants in comparison to control (not UV-A-treated) plants (Figure 2b).

### 3.2. Photosynthetic Apparatus Performance of D. Muscipula Leaves under UV Treatment

To evaluate the level of degradation of photosynthetic apparatus antennae under UV treatment, the contents and the ratio of photosynthetic pigments were estimated. Total chlorophyll content (Chl *a* + *b*) did not change in leaves of UV plants compared to control plants but decreased in recovery plants compared to control and UV plants (Figure 3a), while the Chl *a*/*b* ratio did not change, both in UV and recovery plants relative to control plants (Figure 3a). However, carotenoid content decreased in leaves of UV plants relative to control and in leaves of recovery plants relative to control and UV plants (Figure 3a).

To determine the efficiency of PSII photochemistry, Chl *a* fluorescence was measured. Directly extracted fluorescence parameters (minimum (F_0_), maximum (F_m_) and variable (F_v_) fluorescence) did not change, both in UV and recovery plants compared to control (Figure 3b). Likewise, the maximum quantum yield (F_v_/F_m_) and water-splitting complex activity (F_v_/F_0_) also remained unchanged in these plants. On the other hand, the plastoquinone pool (area) increased only in UV plants in relation to control ones (Figure 3b). The relative variable fluorescence at 2 ms (V_J_) increased in recovery plants and did not change in UV plants compared to control. In turn, the relative variable fluorescence at 30 ms (V_I_) declined in UV plants, while it increased in recovery plants (Figure 3b). Total electron carriers (Sm) increased in UV plants and decreased in recovery plants in comparison to control (Figure 3b). Among the parameters describing yield or flux ratios, the quantum yield for the reduction of end acceptors of PSI per photon absorbed (φ_R0_) and quantum yield for the reduction of end acceptors of PSI per photon absorbed (ρ_R0_) increased in UV plants and decreased in recovery plants in relation to control plants (Figure 3b). Moreover, the quantum yield for electron transport (φ_Eo_) and probability (at time 0) that trapped excitons move an electron into the electron transport chain beyond (ψ_Eo_) decreased in recovery plants compared to control and UV plants (Figure 3b). Parameters describing specific and phenomenological fluxes or activities per reaction center (RC) or cross sections (CS) remained unchanged in UV plants compared to control plants (Figure 3b). These parameters also did not change in recovery plants, with the exceptions of electron transport flux both per RC (ET_o_/RC) and CS (ET_o_/CS_o_), which decreased in these plants compared to control and UV plants (Figure 3b).

To verify the photosynthesis efficiency of examined plants, gas exchange was measured using infrared. These measurements enabled the determination of actual photosynthesis efficiency (Pn) (at 100 μmol quanta·m^−2^·s^−1^), stomatal conductance (Gs), transpiration (E) and intercellular CO_2_ concentration (Ci). Net photosynthesis increased both in UV and recovery plants compared to control (Figure 3c). An increase in Gs was also noted in UV and recovery plants relative to the control (Figure 3c). Transpiration did not change in both UV and recovery plants relative to the control (Figure 3c), whereas Ci increased in UV plants compared to control and recovery plants (Figure 3c). In addition, in low light intensity (0–50 μmol·m^−2^·s^−1^) and in the range of light intensity between moderate (100 μmol quanta m^−2^·s^−1^) and high (2000 μmol quanta·m^−2^·s^−1^), leaf photosynthesis efficiency of UV plants was significantly higher than in control and recovery plants (Figure 3d), while the leaf photosynthesis efficiency of recovery plants did not change in relation to control in low and moderate light intensity (0–1000 μmol quanta m^−2^·s^−1^) but decreased in high light intensity (2000 μmol quanta·m^−2^·s^−1^) (Figure 3d).

To investigate the impact of UV light on specific protein components of the photosynthetic apparatus, SDS-PAGE and the immunoblotting of chosen elements were performed. The content of PsbA (D1), a core component of PSII, did not change after UV treatment (Figure 4a). Similarly, the contents of the core antennas of PSII, PsbB (CP47) and PsbC (CP43), remained the same, both in UV and recovery plants in relation to control (Figure 4b,c), whereas the content of PsbO, an extrinistic subunit of oxygen-evolving complex (OEC), decreased significantly in recovery plants compared to control (Figure 4d).

### 3.3. Sugar Content in D. Muscipula Leaves under UV-A treatment

One of the compounds responsible for maintaining the overall structure and growth of plants is sugar [57]. There were no changes in soluble sugar content in the leaves of both UV plants and recovery plants (Figure 5). Similarly, the content of insoluble sugars did not change significantly in the leaves of UV and recovery plants in relation to control (Figure 5). However, insoluble sugar content was lower in UV plants than in recovery plants (Figure 5).

### 3.4. Plasma Membranes of D. Muscipula Leaves under UV treatment

The content of malondialdehyde (MDA)—a product of lipid peroxidation—was used to evaluate the integrity of cell membranes of *D. muscipula* leaves [58]. MDA content increased in the leaves of recovery plants compared to both control and UV plants (Figure 6).

### 3.5. Reaction of Antioxidant System of D. Muscipula Leaves to UV-A treatment

#### 3.5.1. Enzymatic Antioxidants Activity and Content

Plants, to prevent the overaccumulation of ROS under stress, also developed protective strategies involving enzymatic antioxidants [59,60]. The most common antioxidant enzymes are superoxide dismutase (SOD), catalase (CAT), ascorbic peroxidases (APXs), and other peroxidases [61]. The activities of SOD, CAT and guaiacol peroxidase (POD) were significantly higher in the leaves of UV plants compared to those of control and recovery plants (Figure 7a). In the leaves of recovery plants, the activities of these enzymes were at the same level as in the leaves of control plants (Figure 7a). Among ascorbic peroxidases, two isoforms were identified: thylakoid (t-APX) and peroxisomal (p-APX) (Figure 7b). t-APX content increased in the leaves of recovery plants relative to control and UV plants (Figure 7b), while there was an increase in p-APX content in both the leaves of UV and recovery plants relative to the control, with the increase in recovery plants also being significantly greater compared to UV plants (Figure 7b).

#### 3.5.2. Low-Molecular-Weight Antioxidant Content

Low-molecular-weight antioxidants include phenolic compounds, among others [61]. Phenolic compounds act as ROS scavengers in plant tissues and exhibit shielding properties [62]. The phenolic compound contents were evaluated to assess the ability of UV-A to induce the accumulation of these compounds in *D. muscipula*. Total phenolic compound contents increased significantly in the leaves of recovery plants compared to control plants (Figure 8a). The same changes were also noted for flavonol contents (Figure 8b). In contrast, the contents of phenylpropanoids (cinnamic acid derivatives) did not change in both the leaves of UV plants and recovery plants (Figure 8b). On the other hand, the contents of anthocyanins increased significantly in UV plants and decreased in recovery plants in relation to control plants (Figure 8b). Of the individual phenolic compounds tested, only plumbagin content increased in plants immediately after UV-A treatment (UV plants), relative to both control and recovery plants (Figure 8c). On the other hand, the contents of hyperoside and caffeic acid increased statistically significantly in the leaves of recovery plants compared to control and UV plants (Figure 8c). Mirecithin and ellagic acid contents did not change, regardless of the treatment (Figure 8c). Another low-molecular-weight antioxidant we analyzed is glutathione. Glutathione is involved in various biosynthetic pathways, detoxification, redox homeostasis and signalling functions [63]. Total glutathione (reduced glutathione (GSH) and glutathione disulfide (GSSG)) content increased significantly only in UV plants, while in recovery plants it decreased to the same level as in control plants (Figure 8d).

## 4. Discussion

Although UV-A radiation represents the vast majority of UV radiation reaching the Earth’s surface, studies examining its effects on plants are still scarce [7]. In addition, UV-A radiation, characterized by longer wavelengths (315–400 nm), can penetrate to much deeper target sites in plants than UV-B or UV-C [1]. The implication is that despite their lower reactivity, high levels of UV-A radiation reaching tissues can disrupt various processes and structures in plants [7]. UV-A radiation reaches, among others, mesophyll, where it affects the photosynthetic apparatus, including photosynthetic pigments [18]. The reduction in photosynthetic pigment content is one of the protection mechanisms to prevent damage to photosynthetic apparatus [7]. This mechanism is designed to reduce the size of photosynthetic antennas to limit excessive energy flow to PSII [18]. The key photosynthetic pigments, constituting structural elements of photosynthetic antennas, are chlorophyll *a* and chlorophyll *b*. Chlorophyll *a* constitutes reaction centers (RC) of PSII and PSI and chlorophyll *b* is found only in LHCs [26,64]. LHCs absorb photosynthetically active radiation (PAR) and transfer absorbed energy to RCs, enabling photosynthesis [65]. A few studies have demonstrated that UV-A radiation reduces both Chl *a* and Chl *b* contents [6,66]. Our study showed that a 24 h exposition to UV-A treatment (UV plants) did not affect chlorophyll content in *D. muscipula* leaves. However, the total Chl content declined in the leaves of recovery plants, while the Chl *a/b* ratio remained unchanged both in UV and recovery plants. This indicates that the plants were able to cope with the excess absorbed energy for 24 h exposition to UV-A. The unaffected Chl *a/b* ratio indicates a simultaneous decrease in Chl *a* and Chl *b* contents, and consequently the amount of RCs and the size of photosynthetic antennas. Apart from chlorophyll, carotenoids perform a crucial role in photosynthesis [67]. In photosynthetic organisms, carotenoids have two main functions. The first is light absorption and energy transfer to RCs. The second is the protection of photosynthetic apparatus against ROS generated in chloroplasts as an effect of stress [67]. ROS, in particular, singlet oxygen, may lead to the oxidation and decomposition of carotenoids, resulting in the formation of aldehydes, ketones, endoperoxidases and lactones. These compounds act as signaling molecules causing changes in gene expression, leading to the acclimatization to stress conditions [68]. Studies show various responses of carotenoids to UV-A radiation. On the one hand, treatment with UV-A caused a reduction in carotenoid content in some annual desert plants [66]. On the other hand, in *Lathyrus sativus* plants cultivated under UV-A-supplemented light, the content of carotenoids did not change [18]. In the present study, the accumulation of carotenoids decreased in *D. muscipula* leaves, both in UV and recovery plants. The reduction in carotenoid content suggests their degradation to photoprotect photosynthetic apparatus [18], their transformation to signaling molecules [68], and disruptions of their biosynthesis pathway [69].

Since shorter-wavelength electromagnetic radiation (UV-A: 315–400 nm) is composed of higher-energy photons [70], UV radiation provides much more energy to plants. Such excess energy causes direct damage to the oxygen-evolving complex (OEC) as a result of the impairment of the manganese cluster [5,18], plastoquinone pool, RuBisCo and Atpase [7,8]. Moreover, the absorption of excess energy by light-harvesting complexes (LHC) leads to the overexitation of PSII RC, which generates ROS, responsible for indirect UV-A damage to PSII proteins and other structures of the photosynthetic apparatus and the cell [4,71,72]. Disorders in the functioning of the OEC and PSII lead to disturbances in linear electron transport, resulting in the danger of PSI photooxidation [7]. A similar effect may be induced by the overexitation of external antennas of PSI (LHCI) by UV-A radiation. Although, PSI sensitivity to UV is considerably lower than PSII [4,71,72].

A fast Chl *a* fluorescence kinetics assay was used to evaluate the response of *D. muscipula* photosynthetic apparatus to UV-A treatment. This measurement is a useful, non-invasive, and reliable method for estimating the effects of stress on the photosynthetic apparatus [73,74]. The measurement showed that, in UV plants, although they received an increased amount of energy (according to Planck’s constant) [70], the functioning of PSII reaction centers (RC) was not affected, both at the functional (no changes in ET_o_/CS, ET_o_/CR, V_J_ and F_0_ parameters [18,75,76,77]) and structural levels (no changes in the amount of D1 protein). PSII RC was not damaged because there was no limitation on the donor side, both functionally (no change in F_V_/F_0_ parameter [77]) and structurally (no change in the amount of PsbO). Moreover, there was also no limitation on the acceptor side of PSII. In UV plants, we observed an increased plastoquinone (PQ) pool (area), increased rapidly reducing PQ in the total PQ pool (VI) and an increase in other membrane transporters (Sm) in relation to control plants. No limitation on the acceptor and donor sides of PSII results in the increased efficiency of linear electron transport, whereas the increased rate of electron flow generates a huge risk of uncontrolled formation of ROS [77,78]. An increased supply of electrons on the donor side was noted in Venus flytrap UV plants due to an increased pool of glutathione participating in the ascorbate–glutathione cycle [79], supplying electrons directly to tyrosine Z (Tyr_Z_) and thus protecting free radical formation within PSII. The increased risk of ROS generation is also evidenced by the increased activity of antioxidant enzymes (SOD, CAT, POD, APX), whose function is to scavenge free radicals that occur during both linear and cyclic electron transport (including water–water cycle, Mehler reaction, and chlororespiration) [77]. However, no limitation of electron transport enabled very efficient CO_2_ carboxylation within a wide range of light intensities (20–2000 μmol quanta·m^−2^·s^−1^). More efficient carboxylation also resulted from enhanced stomatal conductance (Gs) [20,21] that significantly increased intracellular CO_2_ concentration (Ci). More efficient carboxylation could have resulted from the vicinity of the photosynthetic apparatus, including the increased accumulation of anthocyanins, which convert absorbed UV radiation to long-wave radiation in the PAR range [19,20], resulting in an increased amount of energy available for carboxylation. Moreover, the protective effect on the function of the photosynthetic apparatus could have increased the accumulation of plumbagin, which acts as an antioxidant and/or screening compound [24] as it can undergo redox cycling due to its chemical structure [32]. The effectiveness of the previously described protective mechanisms against ROS, in addition to efficient carboxylation, is also confirmed by the lack of damage to membrane structures (no change in MDA content) that often accompanies disturbances in redox homeostasis under stress conditions [58].

In contrast, the measurement of Chl *a* fluorescence in recovery plants showed a decrease in the number of active RC PSII among all RC PSII (decrease in F_0_; increase in V_J_ [18,74,76,77]). At the same time, a decrease in the efficiency of electron transport from PSII-active RCs was observed (decrease in ET_o_/RC; ET_o_/CS [76,77]). However, the observed changes were not due to a limitation on the donor side of PSII. Despite a structural change in OEC—a decrease in the amount of its main component (PsbO)—no malfunction of OEC was observed (no change in F_V_/F_0_ [44]). Intensive photochemical reactions can lead to the formation of a high number of slowly reducing (inactive) RC PSIIα super-complexes [80]. At the same time, a decrease in the size of the antennas was observed (decrease in Total Chl), reducing the amount of energy provided to PSII RC. On the other hand, reduced electron transport efficiency was related to the limitation on the acceptor side of PSII. This limitation was caused by a reduction in electron transport between the Q_A_ and Q_B_ sites (φEo decrease). In addition, the limitation could result from a decrease in the PQ pool (area), especially its fast-reducing fraction (VI) and also other membrane transporters (Sm) [74,76]. All of these limitations possibly resulted from direct UV-A effects on electron binding sites (Q_A_ and Q_B_) and membrane PQ [5,6]. Moreover, in recovery plants, the limitation of electron transport efficiency was also associated with a limitation on the acceptor side of PSI (φRo, δRo, and ρRo decrease). The observed changes in the structure and function of photosynthetic apparatus were accompanied by a decrease in Ci despite no changes in stomatal conductance (Gs). A reduction in linear electron transport, coupled with a decrease in CO_2_ availability, resulted in an increased contribution of alternative electron transport pathways (t-APX content increase) as well as an elevation of photorespiration (p-APX content increase). As a result, a decrease in carboxylation was observed at high radiation intensities. The disruption of electron transport resulted in the enhanced generation of ROS, as evidenced by an increase in the level of membrane lipid peroxidation (MDA content increase). Following the disruption of the effective functioning of the photosynthetic apparatus in recovery plants, an increase in the synthesis of selected secondary metabolism compounds (total phenols, flavonols, caffeic acid and hyperoside), associated with the structural remodeling of the cell wall [81], was observed to reduce the amount of radiation reaching organelles, including chloroplasts.

## 5. Conclusions

The research presented in this paper indicates that *D. muscipula* plants underwent two phases of adjustment to increased UV-A radiation. The first—observed in UV plants—was a regulatory phase. The observed adjustments in the performance of the photosynthetic apparatus were related to the exploitation of available mechanisms to prevent the over-reduction of RC PSII: (1) an increase in electron transport efficiency on the donor side; (2) the efficient transport of electrons between photosystems; (3) the efficient exploitation of these electrons on PSI’s acceptor side, especially to efficient dark phase; and (4) the usage of alternative electron transport pathways. In addition, UV plants increased the accumulation of plumbagin as a potential component of a protective mechanism against the disruption of redox homeostasis. The second—observed in recovery plants—was an acclimatization process. In these plants, the regulatory phase was run down, resulting in a decrease in the photosynthesis efficiency, which implied the initiation of the acclimatization process associated with: (1) the activation of alternative electron transport cycles and (2) reduction in the amount of absorbed energy by the diminution of antennas and remodeling cell wall structures to ensure passive protection against excessive radiation reaching the photosynthetic apparatus.

## Figures and Tables

**Figure 1 cells-11-03030-f001:**
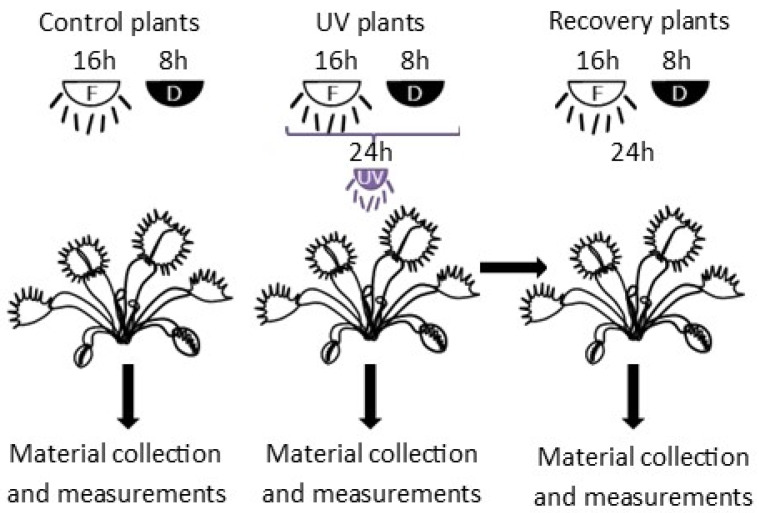
Scheme of experiment. F—fluorescent light (290 μmol·m^−2^ s^−1^ PPFD); D—dark; UV—UV radiation (50 μmol·m^−2^·s^−1^ PPFD).

**Figure 2 cells-11-03030-f002:**
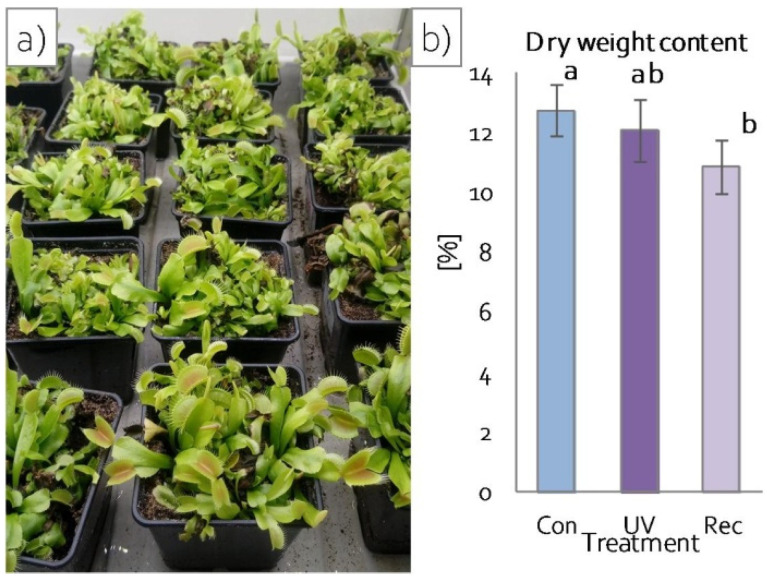
Growth of *D. muscipula* plants under UV-A treatment: (**a**) plant morphology; (**b**) dry weight content; Con—control plants; UV—UV plants; Rec—recovery plants; different letters—statistically significant differences at *p* ≤ 0.05; *n* = 5.

**Figure 3 cells-11-03030-f003:**
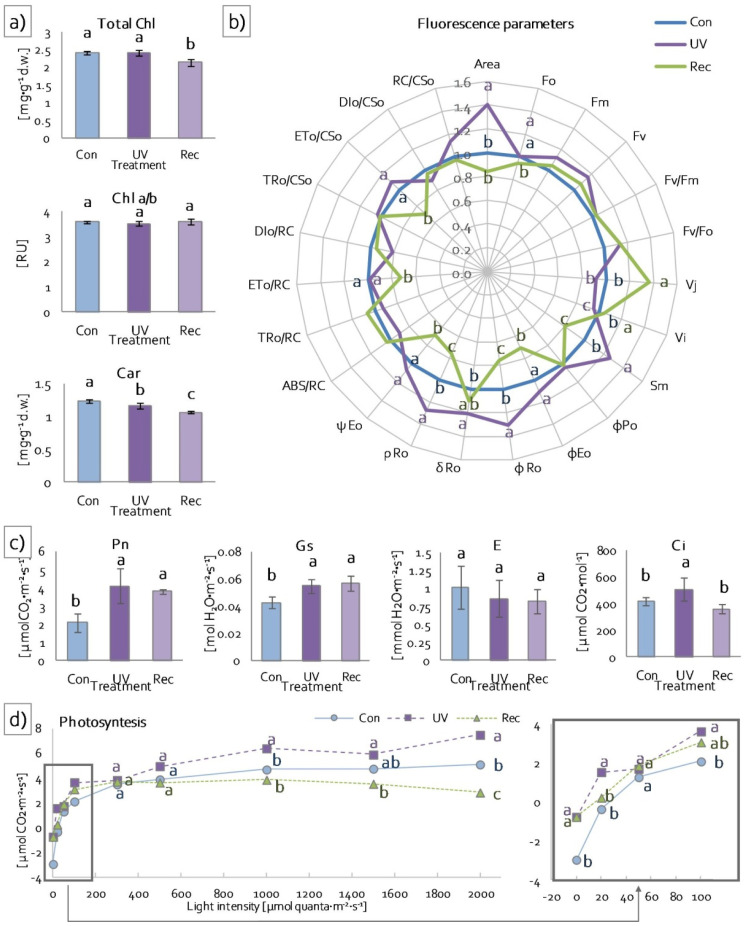
Response of *D. muscipula* photosynthetic apparatus to UV-A treatment: (**a**) photosynthetic pigment content and ratio (*n* = 5); (**b**) structural and functional parameters of photosynthetic apparatus (*n* = 10); (**c**) net photosynthesis (Pn), stomatal conductance (Gs), transpiration and intercellular CO_2_ concentration (Ci) at 100 μmol quanta·m^−2^·s^−1^ (*n* = 3); (**d**) photosynthesis efficiency (*n* = 3); Chl *a + b*—total chlorophylls, Chl *a*—chlorophyll *a*, Chl *b*—chlorophyll *b*, Car—carotenoids; Con—control plants; UV—UV plants; Rec—recovery plants; RU—relative units; all of the values in (**b**) are expressed relative to the control (set as 1); abbreviations—see Section 2.3.2.; different letters—statistically significant differences at *p* ≤ 0.05.

**Figure 4 cells-11-03030-f004:**
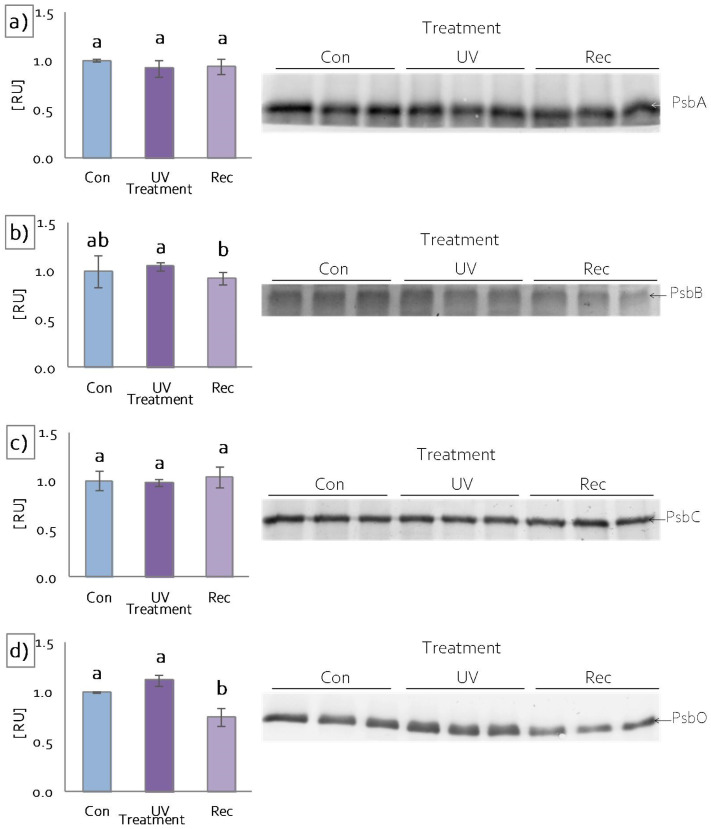
Response of *D. muscipula* photosynthetic apparatus to UV-A treatment. Content of (**a**) D1 protein of PSII (PsbA), (**b**) CP47 protein of PSII (PsbB), (**c**) CP43 protein of PSII (PsbC) and (**d**) oxygen-evolving enhancer protein 1 (PsbO); content of proteins expressed as relative units (RU); Con—control plants; UV—UV plants; Rec—recovery plants; different letters—statistically significant differences at *p* ≤ 0.05; *n* = 3.

**Figure 5 cells-11-03030-f005:**
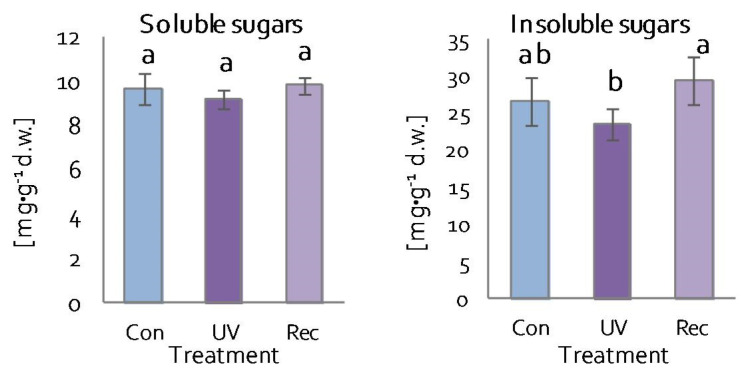
Soluble and insoluble sugar contents in *D. muscipula* leaves under UV-A treatment; Con—control plants; UV—UV plants; Rec—recovery plants; different letters—statistically significant differences at *p* ≤ 0.05; *n* = 5.

**Figure 6 cells-11-03030-f006:**
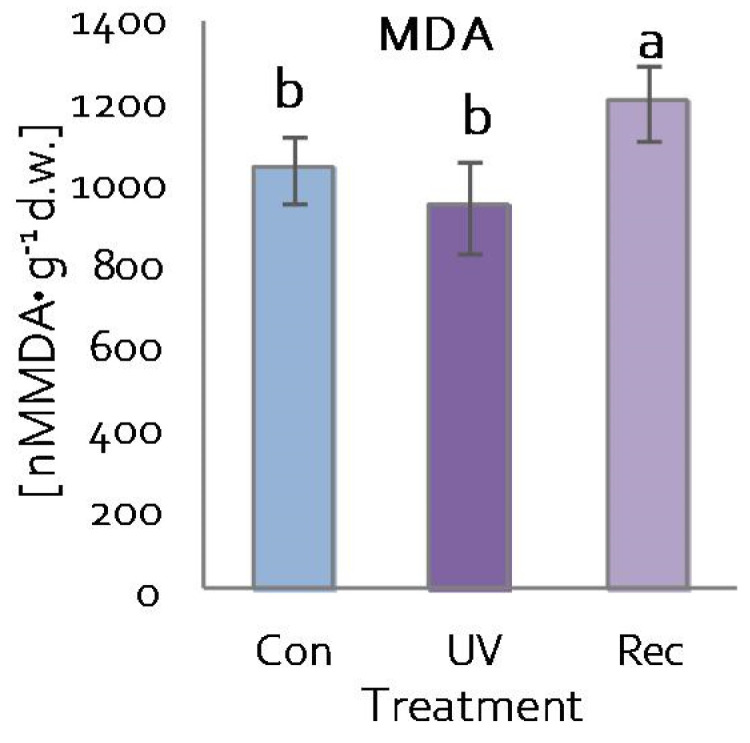
Malondialdehyde (MDA) content in *D. muscipula* leaves under UV-A treatment; Con—control plants; UV—UV plants; Rec—recovery plants; different letters—statistically significant differences at *p* ≤ 0.05; *n* = 5.

**Figure 7 cells-11-03030-f007:**
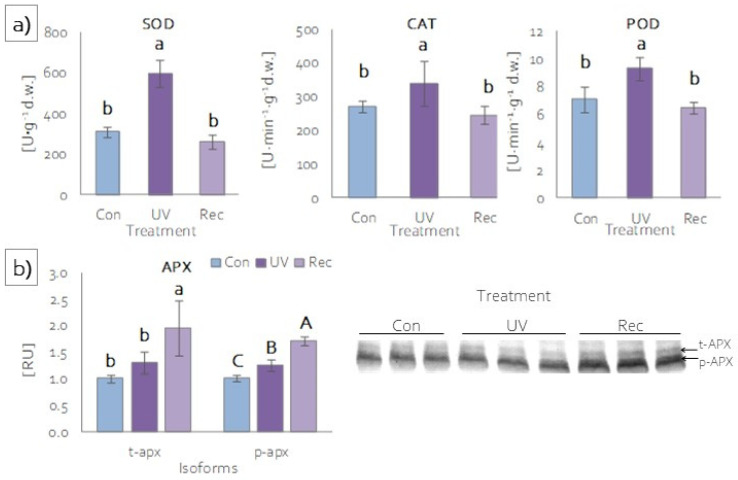
Activity and content of antioxidant enzymes of *D. muscipula* leaves under UV-A treatment: (**a**) activities of superoxide dismutase (SOD), catalase (CAT) and (**c**) guaiacol peroxidase (POD) (*n* = 5); (**b**) content of ascorbic peroxidase isoforms (*n* = 3); content of proteins expressed as relative units (RU); Con—control plants; UV—UV plants; Rec—recovery plants; different letters—statistically significant differences at *p* ≤ 0.05.

**Figure 8 cells-11-03030-f008:**
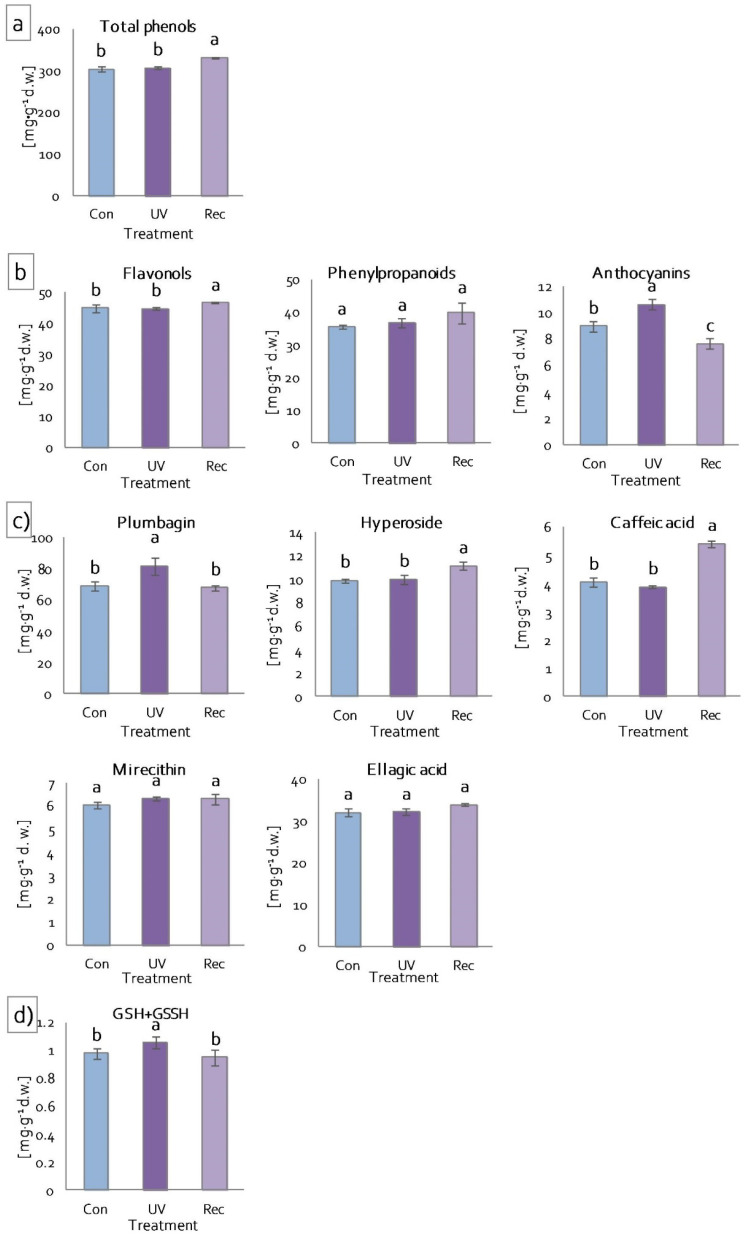
Content of low-molecular-weight antioxidants of *D. muscipula* leaves under UV-A treatment: (**a**) total phenolic compound; (**b**) phenolic compound group; (**c**) specific phenolic compound; (**d**) total glutathione; Con—control plants; UV—UV plants; Rec—recovery plants; different letters—statistically significant differences at *p* ≤ 0.05; *n* = 3.

## Data Availability

Data are contained within this article.

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
