# Peer review of "The Adjustment Strategy of Venus Flytrap Photosynthetic Apparatus to UV-A Radiation"

_cells, 2022, doi:10.3390/cells11193030_

Round 1

Reviewer 1 Report

The manuscript is well written except few typos (line nos. 70 and 85).

The topic looks original to the best of my knowledge and feel this would drive researchers to further take detailed studies from here. This adds on the variable parameters that are associated with the mechanism to build UV-A tolerance. The text is clear and easy to go through the content. Conclusions are OK with respect to the evidences.
The manuscript address the mechanism through which the venus flytrap tolerates UV-A radiation and the study is focused on the analysis with special reference to photosynthetic parameters. The manuscript is relevant and is interesting that it discloses the way the plant tolerates the UV-A radiation.
The authors do address the main question posed.

However, in my personal opinion, adding transcriptome analysis data would have thrown light on complete biological pathway variability associated with UV-A radiation than to study specifically on the photosynthetic traits alone would have added value to the manuscript and possibly give the readers a better understanding.

Author Response

Response to Reviewer 1 Comments

Point 1: The manuscript is well written except few typos (line nos. 70 and 85).

Response 1: We corrected typos (see manuscript)

Point 2: The topic looks original to the best of my knowledge and feel this would drive researchers to further take detailed studies from here. This adds on the variable parameters that are associated with the mechanism to build UV-A tolerance. The text is clear and easy to go through the content. Conclusions are OK with respect to the evidences.
The manuscript address the mechanism through which the venus flytrap tolerates UV-A radiation and the study is focused on the analysis with special reference to photosynthetic parameters. The manuscript is relevant and is interesting that it discloses the way the plant tolerates the UV-A radiation.
The authors do address the main question posed.

Response 2: Thank you for your opinion.

Point 3: However, in my personal opinion, adding transcriptome analysis data would have thrown light on complete biological pathway variability associated with UV-A radiation than to study specifically on the photosynthetic traits alone would have added value to the manuscript and possibly give the readers a better understanding.

Response 3: Thank you for your suggestion. We agree that additional analyses, including transcriptome analysis, would expand our understanding of the effects of UV-A radiation on plant physiology. However, in this paper, in line with the Special Issue theme, we have focused on the action of the photosynthetic apparatus under UV-A stress. Nevertheless, in the future, we plan to expand our studies of UV-A effects on holistic plant physiology.

Reviewer 2 Report

The authors of the manuscript (cells-1915154) examined effects of UV-A light on photosynthetic apparatus of a plant Dionaea muscipula by dry weight, fluorescence parameters, pigment content, and so on. The subject seems to be suitable for the audience of Cells; however, some modifications are required before acceptance for publication.

The authors did not show raw data except for a photo (Fig 1a) and the results of Western Blot. Please provide the raw data to show that the analyses by the authors were properly done.

Section 2.1. The light treatments for three kinds of plants (Con, UV, and Rec) are confusing. It would be better to describe the treatment by using a scheme.

In section 2.1., please explain how and why the authors decide the period of UV-A treatment. 

The abbreviations, Con, UV, and Rec are defined in lines 131-133, but these abbreviations are used only in the figures but not in the text. This will bring confusion to the readers.

The abbreviations, Chl and Car are defined in line 142, although should be done in line 58. These abbreviations are not well used in the text.

Lines 466-467. Did the authors carry out UV-A exposition longer than 24 hours?

Lines 488-490. The word “overexcitation” is ambiguous. In this reviewer’s opinion, the RC absorbing UV-A light has a larger excess vibrational energy, but the amount of RC excited does not increase by UV-A light. Please clearly explain here the authors’ idea on the mechanism generating ROS by UV-A.

Please indicate the wavelength values for “UV-A” (line 34), “its longer wavelength” (line 35), “UV-B” (line 36), “UV-C” (line 36), “longer wavelengths” (line 50), and “longer wavelengths” (line 450).

Author Response

Response to Reviewer 2 Comments

Point 1: The authors did not show raw data except for a photo (Fig 1a) and the results of Western Blot. Please provide the raw data to show that the analyses by the authors were properly done.

Response 1: We added exel file with raw data.

Point 2: Section 2.1. The light treatments for three kinds of plants (Con, UV, and Rec) are confusing. It would be better to describe the treatment by using a scheme.

Response 2: We added a scheme (see manuscript, Fig. 1)

Point 3: In section 2.1., please explain how and why the authors decide the period of UV-A treatment.

Response 3: Our preliminary experiments carried out at intervals of 12, 24 and 36 h showed that a 12h-exposure of Dionanea muscipula to UV-A radiation did not cause any significant changes in the functioning of the photosynthetic apparatus, while 36h-exposure to UV-A caused very serious disorders in its functioning. In contrast, a 24-h UV-A exposure revealed functional plasticity of the D. muscipula photosynthetic apparatus, hence our further research focused on this period.

 Point 4: The abbreviations, Con, UV, and Rec are defined in lines 131-133, but these abbreviations are used only in the figures but not in the text. This will bring confusion to the readers.

Response 4: We changed abbreviations in the manuscript text to more descriptive: control plants, UV plants and recovery plants. We left abbreviations in figures (with their explanations in figure captions).

Point 5: The abbreviations, Chl and Car are defined in line 142, although should be done in line 58. These abbreviations are not well used in the text.

Response 5: We changed as suggested.

Point 6: Lines 466-467. Did the authors carry out UV-A exposition longer than 24 hours?

Response 6: We did not carry out UV-A exposition longer than 24h. We changed the sentence to more relevant.

Point 7: Lines 488-490. The word “overexcitation” is ambiguous. In this reviewer’s opinion, the RC absorbing UV-A light has a larger excess vibrational energy, but the amount of RC excited does not increase by UV-A light. Please clearly explain here the authors’ idea on the mechanism generating ROS by UV-A

Response 7: In the quoted passage, the authors described all the dangers associated with the effects of ionizing radiation on key elements of the photosynthetic apparatus. Regarding the discussed results, indeed, no danger of overexcitation is observed, since there is no limitation in electron transport on both the donor and acceptor sides of PS II.

Point 8: Please indicate the wavelength values for “UV-A” (line 34), “its longer wavelength” (line 35), “UV-B” (line 36), “UV-C” (line 36), “longer wavelengths” (line 50), and “longer wavelengths” (line 450).

Response 8: We indicated the wavelength values (see manuscript).

Round 2

Reviewer 2 Report

The manuscript still contains ambiguity in line 613. In the present sentence, it is not necessarily clear what the authors mean by ”overexcitation”, bringing misreading to readers. Is a chlorophyll molecule in RC exited to a highly excited state, or do photons come to a closed RC? 

“CR” needs to be changed to “RC” in some places.

Author Response

Response to Reviewer 2 comments

Point 1: The manuscript still contains ambiguity in line 613. In the present sentence, it is not necessarily clear what the authors mean by ”overexcitation”, bringing misreading to readers. Is a chlorophyll molecule in RC exited to a highly excited state, or do photons come to a closed RC? .

Response 1: We agree that this sentence can lead to misreading, so we deleted it.

Point 2: “CR” needs to be changed to “RC” in some places.

Response 2: We changed as suggested